# Cannabis and Other Substance Misuse: Implications and Regulations

**DOI:** 10.3390/toxics11090756

**Published:** 2023-09-06

**Authors:** Miski Aghnia Khairinisa, Mohammed Alfaqeeh, Syauqi Nawwar Rafif, Fajar Oktavian Muljono, Michelle Natasha Colin

**Affiliations:** 1Department of Pharmacology and Clinical Pharmacy, Faculty of Pharmacy, Padjadjaran University, Sumedang 45363, Indonesia; syauqi20001@mail.unpad.ac.id (S.N.R.); fajar20008@mail.unpad.ac.id (F.O.M.); michelle20002@mail.unpad.ac.id (M.N.C.); 2Master Program in Clinical Pharmacy, Faculty of Pharmacy, Padjadjaran University, Sumedang 45363, Indonesia; mohammed21001@mail.unpad.ac.id

**Keywords:** cannabis, controlled substances, intoxication, misuse, case report

## Abstract

Abusing controlled substances, including cannabis and various drugs, can result in severe intoxication and even death. Therefore, a comprehensive postmortem analysis is crucial for understanding the underlying causes of such fatalities. This narrative review discusses the characteristics of commonly abused controlled substances, the methodologies employed in postmortem analysis, lethal dosage levels, mechanisms of toxicity, side effects, and existing regulations. The focus centers on seven prevalent groups of controlled substances, namely cannabis, opioids, amphetamine-type stimulants, cocaine, new psychoactive substances, and hallucinogens. These groups have been linked to an increased risk of fatal overdose. Most substances in these groups exert neurotoxic effects by targeting the central nervous system (CNS). Consequently, strict regulation is essential to mitigate the potential harm posed by these substances. To combat abuse, prescribers must adhere to guidelines to ensure their prescribed medications comply with the outlined regulations. Through an enhanced understanding of controlled substance abuse and its consequences, more effective strategies can be developed to reduce its prevalence and associated mortality.

## 1. Introduction

Drug and substance misuse refers to the usage of substances for purposes that are illegal or against medical advice [1]. It has negative consequences for health and can manifest as drug dependence or as one of a variety of other problematic or destructive behaviors. This behavior can result in drug dependence and various social and mental health difficulties. The consequences of substance misuse can be severe, including car accidents, driving under the influence arrests, Domestic violence, sexual harassment, child neglect and abuse, suicide attempts and fatalities, strokes, and overdose deaths [2].

According to the United Nations Office on Drugs and Crime (UNODC) World Drug Report 2022, approximately 284 million persons had used drugs within the previous year, with a majority being men aged between 15 and 64. The report identifies seven groups of drugs that are considered the most harmful, including cannabis, opioids, amphetamine-type stimulants, cocaine, new psychoactive substances, and hallucinogens. Among these, cannabis remains the most frequently used drug globally, with an estimated 209 million users in 2020. Other substances follow in sequence, with opioids having 61 million past-year users, amphetamines with 34 million, cocaine with 21 million, and ecstasy with 20 million estimated users in 2020 [3].

Abusing cannabis and other controlled substances can lead to fatal outcomes due to acute toxicity and severe intoxication, highlighting the need for postmortem analysis to understand the causes of death, assess the mechanisms of intoxication leading to death, and find the amounts of these substances in biological samples [4]. The regulation of cannabis and other substances is guided by the World Health Organization (WHO) Narcotics and Psychotropic Control Guidelines of 1984, which provide global guidelines for their control [5].

This narrative review aims to explore the implications and regulations surrounding the misuse of cannabis and other substances. It will present case reports highlighting substance misuse, discussing the characteristics of these substances, postmortem analysis methodologies, lethal dosage levels, mechanisms of intoxication leading to death, and associated side effects. By providing a comprehensive understanding of the implementation and regulation of cannabis and other substance misuse, this review seeks to contribute valuable insights into this critical public health issue.

## 2. Fatal Outcomes of Substance Abuse

A comprehensive overview of documented cases from existing literature, focusing on instances of drug abuse and misuse that have tragically led to fatal outcomes, is summarized in Table 1.

## 3. Cannabis

The UNODC predicts that in 2020 there will be 209.220 million cannabis users worldwide or 4.12% of the world’s population. Cannabis consumption has increased by 23% during the last ten years. The highest rates of cannabis consumption are observed in North America, where 16.6% of the population uses the drug. Australia and New Zealand follow closely, with each country reporting a 12.14% prevalence of cannabis use [3].

Based on the pharmacological effects, ∆9-THC has neurobehavioral effects mediated by the activation of cannabinoid receptor type 1 (CB1) in the Central Nervous System (CNS). A G protein-linked receptor, CB1, plays a role in several physiological processes, such as appetite, pain perception, mood, and memory [22]. Observational studies indicated that recreational cannabis use may lead to acute poisonings, neurotoxicity, and dangerous outcomes, particularly for children accidentally consuming homemade edibles or concentrated products at home, resulting in intensive care unit admissions and even death, while also being associated with an acceleration of cardiovascular age, irrespective of other factors, suggesting potential adverse effects on the cardiovascular system [23,24,25]. One of the fatal consequences of endocannabinoid system dysregulation is Cannabinoid Hyperemesis Syndrome (CHS). The exact pathophysiological mechanism of CHS is not yet fully understood; however, chronic daily cannabis use has the potential to desensitize or suppress CB1 receptors, which would render cannabinoids ineffective as an antiemetic [26,27]. The development of CHS may be associated with an increase in Transient Receptor Potential Vanilloid 1 (TRPV1) due to chronic stress, as shown in Figure 1.

The primary route of THC delivery is through smoking, accounting for approximately 20% to 70% of THC intake. Smoking a cannabis cigarette containing 500 to 1000 mg of cannabis can deliver a THC dose of 0.2–4.4 mg, while the pharmacological effects of cannabis typically require a dose of 2–22 mg. THC concentrations in the brain generally represent only ∼1% of the administered dose and are typically equivalent to 2–44 μg [28]. Doses > 7.5 mg/m^2^ inhaled in adults have been reported to cause more severe symptoms, such as hypotension, respiratory depression, and ataxia. Long-term cannabis consumption can result in cyclic hyperemesis, behavioral problems, and bronchospasm due to Inhalation [29]. Although there is no specific data on the toxic concentration of THC in humans, high doses of cannabis consumption are known to cause organ damage.

In a case report by Nourbakhsh et al. (2018), a deceased individual had a history of nausea and vomiting two days before death. The individual had visited the emergency room, where symptomatic therapy was provided, but it was unclear whether their CHS had improved or worsened. Low plasma glucose levels (34 mg/dL) discovered by biochemical testing following cardiac arrest in the emergency room indicated food intolerance brought on by irreversible CHS. With high urea (80.95 mg/L) and creatinine (3.41 mg/dL) concentrations and a urea/creatinine ratio greater than 100, toxicological analysis of vitreous samples revealed hyponatremic-hyperchloremic dehydration and a prerenal etiology of acute kidney failure. Muscle hemorrhage near the tongue tip suggested seizure activity before death, which could be attributed to hypoglycemia and/or hyponatremia [6].

Acute kidney injury (AKI) is a common consequence of CHS. The clinical characteristics of CHS can be classified into three phases: the prodromal phase, the hyperemesis phase, and the recovery phase. The prodromal phase can last for several months or even years and is often accompanied by nausea and abdominal pain. Patients commonly express a fear of vomiting. The hyperemesis phase, characterized by recurrent episodes of intense nausea and vomiting lasting 24 to 48 h, can lead to AKI, dehydration, and electrolyte imbalances associated with dehydration [30].

On the other hand, the percentage risk of death due to cannabis toxicity is very small and becomes large due to external incidents or special conditions that trigger death. THC is a partial agonist for CB1 receptors [31,32]. However, cannabis toxicity can be fatal because it can trigger traumatic brain injury due to loss of consciousness [32]. In addition, cannabis toxicity will be fatal if it occurs in individuals with a history of heart disease, as it can lead to heightened sympathetic activity, and reduced parasympathetic activity, resulting in tachycardia and an elevation in cardiac output even at low or moderate doses [33]. Therefore, it is necessary to consider the impact of addiction to cannabis consumption, which is fatal to causes death [34].

## 4. Amphetamine-Type Stimulant

Amphetamine-Type Stimulant (ATS) is a class of drugs regulated by the 1971 Convention on Psychotropic Substances. It is made up of synthetic stimulants. This group includes amphetamine, methamphetamine, 3,4-methylenedioxy-methylamphetamine (MDMA and ecstasy), and its analogs. Methamphetamine (MA) is the second most abused illegal drug in the world, following cannabis, and its user population almost surpasses the combined number of heroin and cocaine users. East and Southeast Asia have the highest number of MA users, accounting for about two-thirds of the global user population, followed by the Americas, particularly the United States and Northern Mexico, where approximately one-fifth of MA users reside. In 2020, 34.080 million people used MA globally, representing 0.68% of the total global population. In Europe, MA is the second most commonly used stimulant after cocaine [35].

The main administration routes of MA overexposure are intravenous injection, intranasal insufflation, smoking, and ingestion. Another route that has been reported is the transrectal route, one of the cases described by Takasu, et al. (2021) [8]. The case explained that a deceased individual with a blood MA level of 9.44 μg/mL was considered fatal. In another case, McIntyre et al. (2013) reported methamphetamine levels of 13 mg/L in peripheral blood, leading to pulmonary edema and congestion. No other cause of death was identified, and acute MA intoxication was determined to cause death [9]. Various studies have reported the lethal blood MA level, which according to the Winek criteria, is >10 μg/mL [36]. The lethal MA blood level can range from 1.4 to 13 μg/mL [37]. The reported therapeutic doses, according to studies, are 0.062–0.291 mg/L (30 mg oral dose) and 0.132 mg/L (0.50 mg/kg single intravenous dose) [9]. MA activates the neurotransmitter system, which increases body temperature, blood pressure, pulse rate, cutaneous vasoconstriction, and respiratory rate. High doses of MA continue to be abused, resulting in severe neurotoxicity, cardiovascular dysfunction, pulmonary disease, and several other diseases [38].

The use of amphetamine or its analogs poses significant dangers due to the potential discrepancies between self-reported substance use and analytically confirmed results, raising concerns about the accuracy of identifying these substances and the potential for polydrug use, putting users at greater risk of acute recreational drug toxicity [39]. A study conducted in Najran region, Saudi Arabia mentioned that the extensive use of Amphetamine and its analogs poses a significant threat to public health, as they account for a substantial portion of acute intoxication cases and contribute to severe morbidity and mortality [40]. A retrospective observational cohort study explained the potential dangers of using synthetic cathinone, an analog of amphetamine, which is associated with a higher risk of life-threatening conditions such as tachycardia, hyperthermia, and rhabdomyolysis [41]. Additionally, another retrospective observational study underscored the significant risk of abuse/misuse with immediate-release dextroamphetamine/amphetamine analogs compared to lisdexamfetamine and dextroamphetamine/amphetamine formulations, making them potentially more dangerous for individuals seeking non-therapeutic use [42]. These findings raise concerns about the health risks posed by the abuse potential of certain amphetamine-based medications.

## 5. Opioids

Opioids are a class of drugs used for reducing severe pain. This class includes both illegal drugs such as heroin and synthetic opioids such as fentanyl, as well as prescription pain relievers such as codeine, morphine, and tramadol, among others. According to data compiled by UNODC, in 2020, approximately 61,290 million individuals worldwide, representing 1.2% of the global population, consumed opioids. Half of the opioid users reside in South Asia and Southwest Asia. Opioids are considered the most lethal drug group and are responsible for two-thirds of deaths directly related to drugs, primarily due to overdoses [3].

There are 3 types of opioid receptors, namely µ (MOP), δ (DOP), and κ (KOP). Among these receptors, MOP receptors are the primary target for severe pain relief in opioid therapy [43]. Fentanyl is a synthetic opioid with a strong affinity for MOP, making it a strong analgesic about 50 to 100 times stronger than morphine. Fentanyl is well-known in the form of a transdermal patch which is very comfortable for patients with acute and chronic pain [44]. Fentanyl poisoning can lead to Opioid-Induced Respiratory Depression (OIRD), characterized by a significant decrease in respiratory frequency and regularity, originating from the medullary preBötzinger complex (preBötC) [45,46]. A case reported by Karen L. Woodall et al. described the death of a 42-year-old male who was found with evidence of drug paraphernalia and numerous prescription bottles. Postmortem examination revealed pieces of plastic from a fentanyl patch in his mouth and posterior oropharynx, but not obstructing the airway. Toxicological analysis indicated a high fentanyl concentration in his heart blood, along with the presence of other drugs. The cause of death was attributed to a fentanyl overdose [47].

Tramadol, or O-desmethyl-tramadol (its active metabolite), is an opioid with a high affinity for MOP. It reduces pain by preventing serotonin and norepinephrine from reuptaking their respective amounts in the brain. The therapeutic dose range for a single dose of tramadol is typically 50–100 mg every 4–6 h, with a maximum recommended dose of 400 mg/day. Tramadol toxicity can affect various organ systems, including the CNS (causing seizures, CNS depression, and low-grade coma), the cardiovascular system (resulting in symptoms ranging from palpitations to life-threatening complications such as cardiopulmonary arrest), and may lead to conditions such as rhabdomyolysis and serotonin syndrome [13,48]. A case reported by Koen De Decker et al. mentioned the death of a 28-year-old Caucasian man who had been treated with tramadol for vague abdominal complaints. The patient ingested a benzodiazepine and snored heavily at night. He experienced apnea and cardiac arrest in the morning, leading to extreme acidosis and hypoglycemia. Despite aggressive resuscitation efforts and supportive therapies, he developed acute hepatic and renal failure, with a liver biopsy revealing steatosis and centrolobular necrosis. Clinical toxicological screening ruled out the presence of other toxic drugs, and GC-MS analysis showed only the presence of tramadol [49].

Morphine is a classic opioid class that acts primarily on MOP receptors within the CNS and the Peripheral Nervous System (PNS). Common side effects that occur when consuming morphine are CNS depression, nausea, vomiting, and urinary retention. A morphine overdose can lead to the accumulation of morphine in the brain, resulting in respiratory depression and death [50]. A case reported by Shepard Siegel and Delbert W. Ellsworth showed a patient who, unfortunately, passed away while receiving morphine injections for an extended period due to a severe medical condition. The morphine injections were given four times a day for four weeks, with a gradual increase in dosage. The patient’s environment was typically dimly lit with hospital-type apparatus, where the morphine injections were administered without issues. However, on the day of the overdose, the patient received the injection in a brightly lit living room, which led to an atypical reaction, including small pupils and shallow breathing. The physician suspected a morphine overdose based on these symptoms, and the patient eventually passed away [51].

These case reports emphasize the dangers of opioid misuse, the need for caution and proper administration, the importance of recognizing potential drug interactions, and the significance of environmental factors in opioid administration, providing valuable insights for medical professionals to enhance patient safety and guide appropriate opioid prescriptions. In light of these dangers, it becomes essential for medical professionals to be aware of the lethal doses of different agents within the opioids group, as summarized in Table 2.

## 6. Cocaine

In 2020, it is estimated that approximately 21.5 million individuals, accounting for 0.4% of the world’s population between the ages of 15 and 64, used cocaine at least once in the previous year. Although the prevalence of cocaine use has slightly increased since 2010, the actual number of cocaine users has grown by 32% due to population growth [35].

Misuse of cocaine poses various potentially fatal side effects, including acute myocardial. Ventricular free-wall rupture, a highly lethal complication, occurs in up to 2% of patients facing acute myocardial infarction [61,62]. A recent case reported by Massimiliano Esposito et al. demonstrates a rare complication of acute cocaine poisoning—heart rupture. The study describes an individual found deceased in a drug dealing district with no external injuries. Autopsy findings revealed massive cardiac tamponade and a full-thickness tissue lesion on the posterior wall of the left ventricle, resulting from acute cocaine intoxication [63]. In two of the cases that were studied, the cause of death was speculated to be from cocaine-induced cardiovascular disease. The dose for each case respectively is 2.0 mcg/mL and 1 mg/L in blood samples. These intoxications caused complications that lead to respiratory failure resulting in death [14,15]. This was attributed to the fact that cocaine exerts its influence on thrombosis by triggering platelet activation, enhancing platelet aggregation, elevating plasminogen activator inhibitor activity, and increasing levels of fibrinogen and von Willebrand factor. These actions result in heightened myocardial oxygen demand, coronary artery vasoconstriction, and intensified platelet aggregation, culminating in thrombus formation and potentially leading to myocardial infarction [61,62,64,65,66,67,68].

## 7. New Psychoactive Substances

The term “New Psychoactive Substances” (NPS) refers to compounds of abuse, whether in their pure form as well as in preparations, which may be analogs of illegal substances or newly developed chemicals intended to imitate the psychoactive effects of prohibited substances [69]. NPS can be categorized into 4 primary categories: synthetic stimulants, synthetic cannabinoids, synthetic hallucinogens, and synthetic depressants. Mephedrone and N-ethylpentylone, listed in Table 1 as NPS, belong to the class of synthetic cathinone, indicating the frequent inappropriate use of synthetic stimulants [70]. In 2020, data from 77 nations across all continents indicated the use of NPS within their territories. Among the mentioned NPS, ketamine was reported by 56 nations, and 38 countries mentioned synthetic cannabinoids. Of the 23 nations with available data, 21 stated that 1% or less of their population had used NPS in the previous year. Due to their relatively uncommon usage, NPS is not regulated by the Convention on Psychotropic Substances of 1971 or the Single Convention on Narcotic Drugs of 1961, raising public health concerns [35].

Mephedrone has a similar abuse potential characteristic to MDMA, with a few differences, such as a quicker onset and less long duration of effects, likely due to its rapid elimination half-life [71]. Based on a recent study, mephedrone’s plasma levels in a non-fatal intoxication were 0.15 mg/mL [72]. There have been cases of non-fatal mephedrone poisoning in living people where the plasma concentration of the drug ranged from 0.01 mg/L to 0.74 mg/L, either alone or in combination with other drugs [73]. The median lethal dose (LD50) of mephedrone in the Mephedrone-Induced Neurotoxicity Test in Rats was determined to be 242.72 μM [74]. Mephedrone, a methamphetamine-like substance, has been associated with arrhythmias, vasospasm, atherosclerosis, acute coronary syndrome, sudden cardiac death, and cardiomyopathy [75]. Autopsy findings in cases of mephedrone intoxication have revealed lung damage and brain stem failure. Exposure to mephedrone could also result in a loss of barrier properties and primary Human Brain Microvascular Endothelial Cells (hBMVEC) dysfunction, potentially promoting neurotoxicity through blood-brain barrier (BBB) degradation [76]. In a reported case by Piotr Adamowicz et al., the death of a young male was investigated, who was discovered critically ill with eight small plastic bags of white powder in his jacket pocket. Initial spot tests suggested the presence of 4-bromo-2,5-dimethoxyphenethylamine (2C-B) in the powders, but routine drug screening failed to confirm its presence in blood and vitreous humor samples. Subsequent analysis, utilizing liquid chromatography-tandem mass spectrometry, identified the powders as containing 4-methylmethcathinone (mephedrone) with a purity range of 80.4–87.3%. Detailed toxicological examination revealed fatal mephedrone intoxication, with concentrations of 5.5 mg/mL and 7.1 mg/mL in the blood and vitreous humor, respectively [16].

N-ethylpentylone, a psychomotor stimulant, may have serious negative effects to cause severe cardiovascular and neurological adverse effects. The LD50 of N-ethylpentylone in mice was reported as 240 mg/kg, but the route of administration was not specified [77]. Seizures, muscle spasms, hallucinations, hyperthermia, nausea, vomiting, rhabdomyolysis, renal failure, and arrhythmia are severe side effects linked to the usage of NPS. These effects may be associated with serotonin syndrome, as numerous cathinones have been demonstrated to promote the release of serotonin, dopamine, and norepinephrine from intracellular storage vesicles [78]. Chisom Ikeji et al. reported a distressing case of a previously healthy 21-year-old man who tragically lost his life after ingesting N-ethylpentylone. Post-mortem toxicology testing confirmed drug exposure, revealing N-ethylpentylone in the urine. The patient’s initial presentation included altered mental status, combative behavior, and subsequent cardiac arrest after receiving haloperidol for agitation. Throughout his course, he displayed severe metabolic acidosis, elevated creatinine kinase, acute kidney injury, disseminated intravascular coagulation, cerebral hypoxia, and myoclonus in the lower extremities. Despite aggressive medical interventions, such as hypothermia cooling, intravenous fluids, and continuous renal replacement therapy, the patient’s condition progressively worsened, leading to cardiac arrest and ultimately, his unfortunate demise [79].

## 8. Hallucinogens

Hallucinogens are a diverse group of naturally occurring and synthetic drugs that induce distorted states of consciousness, perception, thinking, and feeling, accompanied by different degrees of auditory or visual hallucinations. The two primary categories of hallucinogens are psychedelic and dissociative. Psychedelic substances, such as psilocybin and lysergic acid diethylamide (LSD), primarily affect 5-hydroxy-tryptamine (5-HT)2A receptors as the target for serotonin. Meanwhile, dissociative drugs work by blocking the action of N-methyl-D-aspartate (NMDA) receptors. Member States implies that hallucinogenic drug use is ranked lower than other substances, which suggests that it is less of a worry than using cannabis and other drugs globally, with an average score of 5.3 throughout 2013–2017 [3].

This review discusses dextromethorphan (DXM) as a dissociative and MDMA, also known as ecstasy, as a psychedelic. MDMA primarily promotes the production of serotonin (5-HT), noradrenaline, and dopamine. Most of MDMA’s typical effects can be defined by the 5-HT system’s activation, which can decrease depressive and anxious sensations, decrease amygdala fear responses, and boost confidence levels [80]. MDMA concentrations in femoral blood that are greater than 600 g/L are considered comatose-fatal, and the highest recorded plasma MDMA concentration in a survivor was 7.72 µg/mL, resulting from a confirmed overdose [81]. DXM, a dextrorphan prodrug, is an antitussive drug for relieving coughs. As a non-competitive antagonist, DXM is well known for its major site of action at NMDA receptors. The antitussive therapeutic dosage (90–120 mg) is far lower than the acute amount of DXM that drug users have reported for dissociative hallucinogenic effects, which is 150–1500 mg. Therapeutic values of DXM vary from 0.005 to 0.06 mg/L in the blood, while documented lethal doses range from 3.3 to 9.5 mg/L in the blood and 31 to 230 mg/kg in the liver [82].

MDMA’s inhibition of the cytochrome-2D6 (CYP2D6) enzyme may increase the toxicity of concurrently used medications, as this cytochrome is involved in the metabolism of several psychoactive substances. Poor metabolizers, individuals with a defective or aberrant CYP2D6 phenotype, may be more susceptible to acute toxicity and side effects [83]. Genetic mutations, specifically on the ryanodine receptor, can lead to malignant hyperthermia with rhabdomyolysis in cases of ecstasy use [84]. Dextrorphan binds to serotonin receptors and, in conditions of overdose, may result in seizures, rigidity of the muscles, instability of the autonomic nervous system, and rhabdomyolysis, which are symptoms of the serotonin syndrome [85].

## 9. Safety and Toxicological Evaluation

Use and misuse of alcohol, nicotine, and illicit drugs, and misuse of prescription drugs cost Americans more than $700 billion a year in increased health care costs, crime, and lost productivity [86]. The misuse of drugs will cause addiction as a chronic, relapsing disorder characterized by compulsive drug seeking. It is considered a brain disorder because it involves functional changes to brain circuits involved in reward, stress, and self-control [87]. Deaths involving synthetic opioids other than methadone (primarily fentanyl) continued to rise, with 70,601 overdose deaths reported. Those involving stimulants, including cocaine or psychostimulants with abuse potential (primarily methamphetamine), continued to increase, with 32,537 overdose deaths in 2021 [88].

Since most patients who overdose on drugs are lethargic or comatose, the history is usually obtained from family, friends, bystanders, and emergency medical service providers. On many occasions at the scene, one may find pills, empty bottles, needles, syringes, and other drug paraphernalia. Other features that one should try and obtain in the history are the amount of drug ingested, congestion, and time of ingestion [89]. Rapid identification of the toxidrome saves time in evaluating and managing a poisoned patient. (Figure 2).

Substance Abuse Evaluations (SAEs) are conducted by medical professionals trained to evaluate individuals suspected of abusing substances. These evaluations include physical examinations, psychological testing, and interviews with family members, friends, coworkers, supervisors, and others familiar with the person being evaluated. The evaluation results are then used to determine whether or not the individual should be referred for treatment. SAEs can also determine the duration to which substance use has impacted their life and the degree of an individual’s drug or alcohol abuse/addiction if there are co-occurring concerns (e.g., depression, anxiety, etc.) [90].

Considering drug overdose or toxicity in a lethargic patient with no other identifiable cause is important. Patients with drug overdose usually undergo several investigations. Plant-derived natural drugs exhibit a rich variety of chemical structures and demonstrate a broad spectrum of activities, spanning a wide range of potential effects [91]. Utilizing technologies such as modern chemical proteomics technology will enhance the comprehension of the clinical toxicity and side effects associated with various drugs, including cannabis. This technique precisely identifies the protein targets affected by these compounds, analyzes intricate interactions between proteins and these compounds, and maps out resulting disruptions in cellular pathways [91].

Drug screens are readily available but often do not change the initial management of straightforward cases. In most cases, a positive drug result will appear even 48 h post-exposure. Moreover, research shows that medication should be the first line of treatment to help people detoxify from drugs. However, detoxification is not the same as treatment and is insufficient to help a person recover. Additionally, medication is usually combined with behavioral therapy or counseling. The therapy should be adjusted to address each patient’s drug use patterns and drug-related medical, mental, and social problems [92].

## 10. Regulation

In regulation, substance abuse, also known as Controlled Substances, pertains to drugs or other substances that are subjected to strict regulation by the government due to their potential for abuse. The Controlled Substances Act (CSA) classifies these substances into five schedules [93]. The following describes each schedule and examples, as outlined in Table 3.

Controlled substances require strict handling from the government. In the United States (US), the agencies entitled to remove, add, or change schedules or other substances are the Drug Enforcement Administration (DEA). In determining the schedule for drugs or other substances must be placed, or whether a substance must be discontinued or rescheduled, certain factors must be considered. These factors include (1) The substance’s actual or potential misuse (2) Clinical proof of the drug’s physiological effects, if available (3) The current level of scientific understanding of the substance; (4) Its utilization trends in the past and present (5) The potential for harm to the public health (6) The likelihood of psychological or physical dependence from the substance (7) Establishing whether the chemical is a direct precursor to one that is already controlled [93]. Controlled Substances regulations regarding distribution or use requirements are presented in Table 4.

Effectively monitoring controlled substances poses significant challenges in distinguishing between legitimate medical use and potential illicit drug abuse. Prescribers face the complex task of differentiating between prescriptions intended for lawful treatment and those that may be misused for illicit purposes. To ensure accurate and appropriate treatment using controlled substances, prescribers need to recognize the signs, symptoms, and treatment approaches for both acute and chronic pain and identify potential signs of substance abuse in patients. Several efforts can be made to maintain the accuracy of treatment with controlled substances based on the patient’s medical record, including: (1) All communications must be recorded (2) Records must be legible and include information about dosage changes, efficacy, and negative effects (3) Describe medication compliance or noncompliance (4) Results of a urine drug test (5) Information about conduct (6) Interactions with family (7) In particular, keep records of any decisions to stop receiving treatment and those made by US states to lessen the abuse, diversion, and overdose of prescription drugs, particularly opioids. This statute demands: (1) Protection from legal action for those who seek assistance during an overdose (2) Control of pain clinics (3) Patient identification before administration (4) Physical examination before prescribing opioids (5) Prescription limits (6) Prohibition of obtaining a controlled substance recipe (7) Tamper-resistant recipe. This can be executed to reduce the incidence of overdose or controlled substance abuse [94,95].

Countries, communities, and populations have different levels of risk, protection, and substance use. In Uruguay and Portugal, the law does not criminalize drug use or possession for personal consumption. Consequently, a prison crisis persists in which ever larger numbers of youths and other vulnerable sectors of society situated at the lowest levels of the drug-trafficking chain are inside the prison system [96,97,98,99,100]. Although countries have enacted decriminalization policies, evidence for or against its effectiveness is lacking [96,97,98,99]. Meanwhile, cocaine use is associated with various negative health outcomes, including infectious diseases such as HIV and hepatitis C, mental health problems, and mortality, particularly among people with cocaine use disorder [101,102]. Given the negative consequences associated with cannabis and cocaine use disorders, it is imperative to identify gaps in treatment to inform policy and lower the region’s disease burden.

Services for preventing and treating substance misuse and substance use disorders have been delivered separately from other mental health and general health care services. Effective integration of prevention, treatment, and recovery services across healthcare systems is key to addressing substance misuse and its consequences. It represents the most promising way to improve treatment access and quality. Many health home and chronic care model practices are now used to manage other diseases and have been extended to include the management of substance use disorders. Moreover, the use of health IT is expanding to support greater communication and collaboration among providers, fostering better integrated and collaborative care while at the same time protecting patient privacy. It also has the potential to expand access to care, extend the workforce, improve care coordination, reach individuals resistant to engaging in traditional treatment settings, and provide outcomes and recovery monitoring [99].

In addition to the role of health workers, the role of the government in creating the criminal justice system also needs to be increased. Not only punishment is given to the perpetrators of drug theft, but alternative therapies must be implemented to improve public health and safety. Drug-abusing offenders, before being released from punishment, must be in good health without the influence of drug addiction, which will make them subject to the same punishment [103].

Compared to the general population, those who commit crimes and are incarcerated report greater lifetime rates of drug use and more dangerous usage behaviors (including injecting). Due to this, the criminal justice system and prisons are crucial settings for drug-related interventions. Cannabis usage or possession is a factor in the bulk of drug law violations reported in most European Union countries. Drug users with problematic usage patterns are more likely to report acquisitive crimes, including robbery, theft, and burglary, that are undertaken to pay for drug use. This latter group, which frequently consists of repeat criminals, can account for a sizable fraction of the prison population.

In prisons around Europe, various drug-related initiatives that have been proven successful in the community have been put into place. Equivalence of services to those offered in the community and continuity of care before and after release from prison are two key criteria for health interventions in prison. Various types of medications and strategies can prove effective during different phases of treatment, aiding individuals struggling with substance abuse in ceasing drug use, adhering to medication regimens, and preventing relapses. The stages of treatment are elaborated in detail in Table 5 [92].

The most widely utilized strategies for treating substance use problems are behavioral therapies. In drug addiction therapy, behavioral therapies assist patients in changing their attitudes and drug-related behaviors. As a result, patients are better equipped to deal with stressful circumstances and different triggers that could result in another relapse. Additionally, behavioral therapy can help patients stay in treatment for longer periods and increase the effectiveness of drugs. The behavioral treatment that is often used is Cognitive-Behavioral Therapy (CBT). Evidence from many trials and quantitative reviews shows that CBT is an effective treatment for drug addiction. Over 1400 individuals were treated in 19 randomized trials (cases) chosen and examined. Results from the various instances revealed that patients were treated with various substances, with those who misused marijuana, cocaine, alcohol, and other opioids making up the majority of patients. In most of these cases, it was noted that CBT treatment methods included relapse prevention, cognitive restructuring, and contingency management. CBT for drug addiction treatment contains four main components: (1) learning how to develop alternative behaviors to substance abuse; (2) methods for managing urges; (3) methods for rejecting drugs; (4) methods for enhancing non-drug-related activities as well as other positive stimuli that may lead to a change in behavior [92,104].

## 11. Conclusions

There were 6 prevalent controlled substances abused, such as cannabis, opioids, amphetamine-type stimulants, cocaine, new psychoactive substances, and hallucinogens. These substances carry a significant risk of fatal overdose, which can result in death. These substances primarily affect the CNS, leading to potential neurotoxic side effects. To mitigate the abuse of these substances, it is important for prescribers, such as doctors, to ensure that the medications they administer adhere to the regulations and guidelines established for their appropriate use. By following these rules and prescribing responsibly, healthcare professionals can contribute to reducing the likelihood of substance abuse and its associated harms.

## Figures and Tables

**Figure 1 toxics-11-00756-f001:**
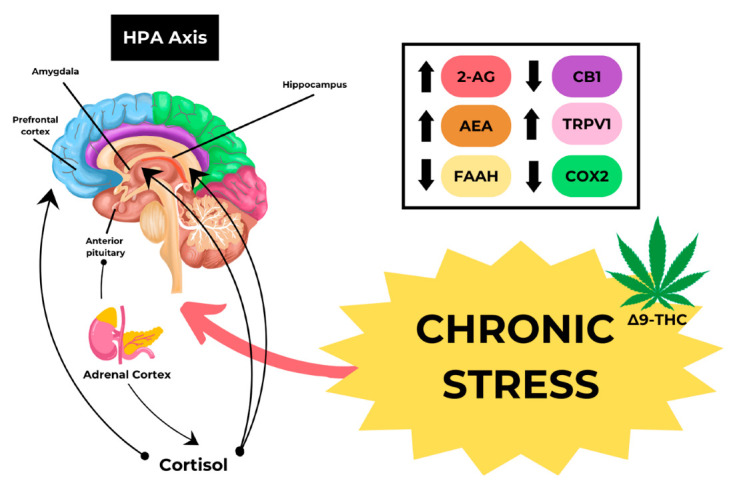
Mechanism of CHS caused by Cannabis.

**Figure 2 toxics-11-00756-f002:**
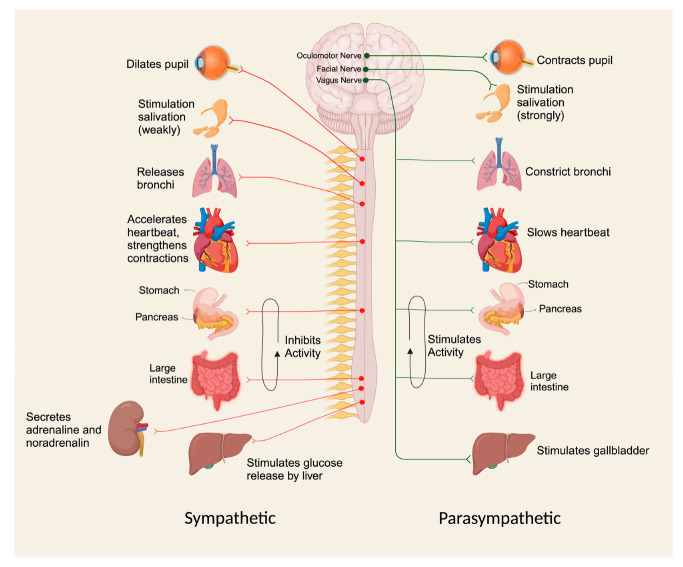
Toxidrome Approach to Poisoning Due to Drugs of Abuse. Created with www.biorender.com, accessed on 10 August 2023.

**Table 1 toxics-11-00756-t001:** Overview of Literature Cases of Drugs Abuse/Misuse resulting in Death.

Reference (Country)	Chemical Substance (Route)	Classification	Type of Biological Sample	Dose		Mechanism of Intoxication	Side Effect
				Sample Concentration	Lethal Dose		
Nourbakhsh et al., 2019 (Canada) [6].	Tetrahydrocannabinol (Inhalation)	Cannabis	Femoral Blood	Delta-9-THC: 7.7 μg/L Carboxy-THC: 24.9 μg/L	not specified	Chronic overstimulation of CB1 cannabinoid receptor activation bypassing the endocannabinoid system will lead to erratic neurotransmitter modulation that can cause toxicity.	Cyclic attacks of nausea and vomiting
Behonick et al., 2014 (Japan) [7]	5F-PB-22 (Inhalation)	Synthetics Cannabinoids	Iliac Blood	1.5 ng/mL	not specified	Chronic overstimulation of CB1 cannabinoid receptors leads to visceral congestion, pulmonary oedema, and pulmonary granulomatous inflammation changes.	Convulsions, tachycardia, respiratory failures, severe agitation, unconsciousness, nausea, vomiting, and death
Takasu et al., 2021 (Japan) [8]	Methamphetamine (Transrectal)	ATS	Blood	9.44 μg/mL	1.4–13 μg/mL	Increasing release of monoamine neurotransmitters such as serotonin, dopamine, and norepinephrine that can lead to Serious neurotoxicity and cardiovascular dysfunction.	Increasing body temperature, respiratory rate, cutaneous vasoconstriction, and blood pressure.
McIntyre et al., 2013 (USA) [9]	Methamphetamine (Oral)	ATS	Peripheral blood	13 μg/mL	1.4–13 μg/mL		Loss consciousness, agonal breathing, but no heartbeat.
Peeters et al., 2022 (Netherland) [10]	Fentanyl (Patch)	Opioid	Subclavian blood	57.9 ng/mL	1.8–81 ng/mL	OIRD originates from medullary preBötzinger complex (preBötC) and is characterized by a significant reduction in the regularity of the inspiratory rhythm.	Respiratory depression, deterioration of the central nervous system, hypothermia, cold, damp skin, floppy muscles, bradycardia, and hypotension
Yonemitsu et al., 2016 (Japan) [11]	Acetyl Fentanyl (Intravenous)	Opioid	Femoral vein blood	153 ng/mL	1.8–81 ng/mL		
Li et al., 2021 (China) [12]	Morphine (Oral)	Opioid	Blood	2.18 µg/mL	0.5 μg/mL	Excessive binding to μ-receptors and inhibition of serotonin reuptake results in decreased CNS function leading to death	CNS depression, tachycardia, seizures, nausea and vomiting.
Gioia et al., 2017 (Italy) [13]	Tramadol (Oral)	Opioid	Femoral Blood	5.8 μg/mL	1.6 μg/mL		
Montoya-Ramirez et al., 2019 (Colombia) [14]	Cocaine (Oral)	Cocaine	Blood, Vitreous humor Nasal smear	2.0 mcg/mL	not specified	Cocaine-induced hypertension which leads to acute respiratory failure	Hypertension, hemorrhagic gastropathy, pulmonary edema
Morentin et al., 2014 (Spain) [15]	Cocaine (Not reported)	Cocaine	Blood	1.0 mg/L	not specified	Cocaine-induced cardiovascular disease	Acute cardiovascular syndrome
Adamowicz et al., 2013 (Germany) [16]	Mephedrone (Oral)	NPS	Blood	5.5 mg/mL	>0.15 mg/mL	Promote BBB degradation, resulting in neurotoxic effect The contraction bands’ necrosis and micro-hemorrhages show that the heart is suffering.	Tachycardia and palpitation, hypertension, dilated pupils, hallucination and psychosis. Arrhythmias, vasospasm, atherosclerosis, acute coronary syndrome, sudden cardiac death, and cardiomyopathy
Anzillotti et al., 2020 (Italy) [17]	Mephedrone (Intramuscular)	NPS	Urine Bile Blood samples	2.0 mg/L (urine) 1.1 mg/L (bile) 0.8–1.0 mg/L (blood)	>0.74 mg/L		
Zawadzki et al., 2020 (Poland) [18]	N-ethylpentylone (Not reported)	NPS	Blood Urine	10.6 μg/mL (blood) 17.6 μg/mL (urine)	not specified	Related to a serotonin syndrome	Psychomotor agitation, confusion, tachycardia, psychosis, inconsistent speech, and cardiac arrest
Politi et al., 2021 (Italy) [19]	MDMA (Oral)	Hallucinogen	Blood Urine	3700 ng/mL (blood) 168,000 ng/mL(urine)	>7.72 μg/mL	Poor metabolizers (abnormal, dysfunctional CYP2D6)	Unconscious, pale and sweaty
Lang et al., 2016 (Germany) [20]	Ecstasy (Oral)	Hallucinogen	Femoral blood	4200 μg/L	600 μg/L	Induce malignant hyperthermia with rhabdomyolysis.	Sweating and hematoma
Shafi et.al., 2016 (Pakistan) [21]	Dextromethorphan (Oral)	Hallucinogen	Whole blood	7.3–41.7 mg/L	3.3–9.5 mg/L	Serotonin syndrome	Euphoria, stupor, hyperexcitability, laughing, nystagmus, mydriasis, nausea, vomiting and diaphoresis

Delta-9-THC: Δ9-Tetrahydrocannabinol, Carboxy-THC: carboxy tetrahydrocannabinol CB1: cannabinoid receptor type 1, 5F-PB-22 (1-pentyfluoro-1H-indole-3-carboxylic acid 8-quinolinyl ester also known as 5F-QUPIC), ATS: Amphetamine-Type Stimulant, OIRD: Opioid-Induced Respiratory Depression, CNS: Central Nervous System, NPS: New Psychoactive Substances, BBB: Blood Brain Barrier, MDMA: 3,4-methylenedioxy-methylamphetamine, CYP2D6: cytochrome-2D6s.

**Table 2 toxics-11-00756-t002:** Lethal doses of various opioid agents.

Opioids	Lethal Dose/s	Reference
Morphine	5 µg/mL	D. H. Eagerton et al. [50]
Codeine	0.5 to 1.0 g	James A. Wright et al. [52]
Oxycodone	1200 mg/L	William P. Tormey [53]
Hydrocodone	0.47–0.38 mg/L	Molina et al. [54]
Fentanyl	1.8 ng/mL–81 ng/mL	Katherine P. Taylor et al.; James R. Gill et al. [55,56]
Heroin	200 mg	Pok P. Rop et al. [57]
Methadone	1 mg/kg	Gaël Dupuy et al. [58]
Buprenorphine	2–20 ng/mL	Bruno Mégarbane et al. [59]
Tramadol	1.6 μg/mL	Samaneh Nakhaee et al. [48]
Hydromorphone	60 ng ⁄mL	Robert Meatherall et al. [60]

**Table 3 toxics-11-00756-t003:** Schedules of Controlled Substances

Schedule	Definitions	Examples
Schedule I	High abuse potential, without a recognized medicinal purpose; medications within this schedule may not be prescribed, dispensed, or administered	Heroin, Marijuana (cannabis), and MDMA (“ecstasy”)
Schedule II	High abuse potential with severe psychological or physical dependence	Morphine, codeine, fentanyl, pentobarbital
Schedule III	Intermediate abuse potential may lead to moderate or low physical dependence or high psychological dependence (i.e., less than Schedule II but more than Schedule IV medications)	Hydrocodone/acetaminophen 5 mg/500 mg or 10 mg/650 mg; codeine in combination with acetaminophen, aspirin, or ibuprofen; ketamine
Schedule IV	Low potential for abuse relative to substances in schedule III.	Alprazolam, clonazepam, diazepam, midazolam, phenobarbital, Tramadol.
Schedule V	Low potential for abuse in comparison to chemicals in schedule IV and largely consist of preparations with trace amounts of specific opioids. These are usually used for analgesic, antidiarrheal, and antitussive uses.	Robitussin AC, Phenergan with codeine

**Table 4 toxics-11-00756-t004:** Controlled Substances Act Requirements Summary [93,94].

Requirements	Schedule II	Schedule III and IV	Schedule V
Registration	Required	Required	Required
Receiving Records	Order Forms (DEA Form-222)	Invoices, Readily Retrievable	Invoices, Readily Retrievable
Prescriptions	Written Prescription (See exceptions *)	Written, Oral, or Fax	Written, Oral, Fax, or Over the Counter **
Refills	No	No more than 5 within 6 months	As authorized when prescription is issued
Distribution Between Registrants	Order Forms (DEA Form-222)	Invoices	Invoices
Security/Storages	Locked Cabinet or other secure storage	Locked Cabinet or other secure storage	Locked Cabinet or other secure storage
Theft or Significant Loss	Report and Complete DEA Form 106	Report and Complete DEA Form 106	Report and Complete DEA Form 106

Note: All records must be maintained for 2 years unless a state requires a longer period. * Emergency prescriptions require a signed follow-up prescription. Exceptions: A facsimile prescription is the original when issued to residents of Long-Term Care Facilities, Hospice Patients, or compounded IV narcotic medications. ** Where authorized by state-controlled substances authority.

**Table 5 toxics-11-00756-t005:** Different Stages of Drug Addiction Treatment

Stages	Description
Treating Withdrawal	When patients first stop taking the drug, they will experience a variety of physical and emotional symptoms, including restlessness and trouble sleeping. Medication and plans need to be made to reduce these symptoms to make it easier to stop taking the drug.
Staying in Treatment	Several medications and mobile applications are used to help the brain gradually adapt to the absence of medication. This medication works slowly and has a calming effect on the patient’s body so they focus on counseling and other psychotherapy they are undergoing.
Preventing Relapse	Science has shown that the most typical trigger for relapse is stress cues associated with drug use (such as people, places, things, and moods). To help patients maintain their recovery, researchers have been working on medicines that interfere with these triggers.

## Data Availability

All data found and analyzed are included in this review article.

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
