# Peer review of "Cannabis and Other Substance Misuse: Implications and Regulations"

_toxics, 2023, doi:10.3390/toxics11090756_

Round 1
Reviewer 1 Report
1、The Materials and Methods section requires a refined analysis of different drugs and their analogs.
2、A detailed overview of the mechanisms of the different classifications of addictive drugs can be provided.
3、Methods of safety and toxicological evaluation of drug overdose can be found in the ratings of the Clinical Pharmacodynamics Index derived from the manuscript entitled 'New opportunities and challenges of natural products research: When target identification meets single-cell multiomics'.
Author Response
Dear Reviewer,
Please look at the attached file as our response to your comments.
Thank you

Reviewer 2 Report
In this review, Khairinisa et al. state that they present case reports that focus on postmortem analysis, lethal dosage, side effects, and causes of death in patients who have misused illicit substances. It is my opinion that reviewing case reports from all over the world to highlight the dangers and risks of drug misuse is innovative, interesting, informative, and helps us understand this public health issue on a global scale. However, it does not appear that the authors stick to this idea and also do not provide a clear description of how the case reports are selected. Below, I have included comments that I hope the authors will consider because I strongly feel that if they remain focused on their novel idea centered around observational studies, this review will make a major contribution to the scientific community and to the public health sector.
1. Materials and Methods, section 2.1-Literature Search, include the total number of articles identified using the search terms for each database. Also state the duplicates found.
2. Materials and Methods, section 2.2-Study selection, include the total of number of articles that fit criteria. Then break that down further to include the total number of articles that fit criteria for each drug.
3. It is not clear how drugs other than cannabis were identified based on the search terms described in Section 2.1. Only the search term, “Cannabis” is included.
4. Did the authors consider related terms in their search, including marijuana for cannabis or heroin, fentanyl, oxycodone for opioid? If so, these need to be included in the Methods section.
5. A clearer description of the focus of this manuscript is required. It appears that the authors intend to provide a review highlighting the dangers of substance misuse to ensure that doctors are adhering to the regulations and guidelines required to reduce misuse. However, the means to this end are not clear. Using case reports, are the authors intending to highlight dangers of illicit substances by focusing on 1) substance use in general, 2) fatal overdose, or 3) toxicity induced by illicit substances? I encourage the authors to consider fatal overdose and toxicity as this will be informative and provide a narrow focus, thus limiting an extensive literature review.
6. For the cannabis section, why not include the following observational studies (PMID: 30676820; PMID: 31615421; PMID: 27821595)? I believe these would provide support for highlighting the dangers associated with recreational use of cannabis. I understand that the authors are complying with a word limit, but the paragraph on lines 117-128 may be cut to fit this pertinent information.
7. For the amphetamine section, the authors should consider including the following observational studies, PMID: 29384873; PMID: 25832473; PMID: 33940992; PMID: 35438590
8. There is not any new information provided in the opioid section that is not already known in the field. The authors may consider providing a table of the lethal doses of various opioids and fill this section with case reports that demonstrate the various modes of death and toxicity that are observed clinically. The authors may consider comparing the effects between different opioids (e.g., heroin, fentanyl, oxycodone) if the case reports are available.
9. For the sedatives section, why not include PMID: 27295906; PMID: 33740697?
10. I encourage the authors to include case reports in the Cocaine Section rather than listing the side effects of using the drug, which are widely known.
11. The NPS section mostly lists the prevalence and pharmacokinetics of this drug class. Based on the paragraph starting on line 52, this review should explore the implications and regulations surrounding drug misuse, which they attempt through the presentation of case reports. As written, the NPS section does not appear to coincide with this focus, but rather provides a description of the pharmacological basis of NPS. I encourage the authors to stick to the described focus of the article. A review of case reports associated with the fatal overdose or toxicity of NPS would be original and more interesting.
12. Inclusion of the Regulations section is not informative. If the authors provide case studies as mentioned for each drug, then the natural conclusion would be the paragraph starting on line 369. This paragraph would provide a seamless end if the authors stick to their focus.
Author Response

(The authors gave the same response as above.)

Reviewer 3 Report
I cannot recommend acceptance of the paper in the present state. This might be possible only after the authors would have submitted a completely revised version.
Literature should be searched more deeply, lot of fatal cases reports are missing regarding NPS. Data are superficial, it would have been better if the authors had focused on one or two classes of compounds.
Line 43: It is not very likely that abusing cannabis will lead to fatal outcome since THC is partial, not full agonist of CB receptors. Please comment your statement.
Line 64: Please explain the expression ‘’mechanism of destruction’’.
Line 227: I propose to delete sections on sedatives.
Line 343: Regulation section should be placed before cases.
Line 350-387: I do not see the point of this section, authors should focus the review on fatal intoxications.
I
Author Response

(The authors gave the same response as above.)

Round 2
Reviewer 1 Report
The author has significantly improved the original manuscript. But, there are minor modifications needed. Here, modern chemical proteomics technology is a rapidly developing tool for identifying new targets of drugs, which has important value in understanding and revealing the clinical toxicity and side effects of Cannabis. So, it is recommended that the author could refer to and cite this literature: Yuyu Zhu#, Zijun Ouyang#, Haojie Du#, Meijing Wang, Jiaojiao Wang, Haiyan Sun, Lingdong Kong, Qiang Xu*, Hongyue Ma*, Yang Sun*. New opportunities and challenges of natural products research: When target identification meets single-cell multiomics. Acta Pharmaceutica Sinica B 2022;12(11): 4011-4039.
Author Response
Dear Reviewer,
We greatly appreciate your comments and suggestions. Please have a look at the attached file to see our response. Thank you
Best regards,
Authors

Reviewer 2 Report
This manuscript is a revised version based on reviewer feedback. The authors were very responsive to the reviewer concerns. In my opinion, the review is almost publishable. However, the authors may consider making the minor corrections that are listed below as often, the authors lose focus and provide unnecessary information that distracts from the purpose. Additionally, thoughts are repeated. The comments below are intended to provide a more concisely written manuscript.
1. I agree, the Methods section may be removed.
2. Section 3. The mechanistic classification of Addictive Drugs should be removed. This is a narrative focused on fatal overdose and toxicity. There is no need to explain the mechanism of addiction.
3. I don’t think L138-146 are necessary and detract from the focus.
4. Consider removing, “MA is a highly addictive CNS stimulant” (L211).
5. Consider removing, “MA is a potent CNS stimulant…” (L219)
6. Consider removing the sentence on L263 (“opioids are severe pain relievers…”)
7. Consider moving the two sentences on L264-266 (“There are 3 types… Among these receptors…”) to L254 inserted between the sentence ending in “..and tramadol, among others” and “According to the data…”
8. Consider removing L318-320 and instead beginning the Cocaine section with the sentence, “In 2020,…
9. Consider removing the sentence on L381. It is unclear why providing an in vitro mechanism of action that is likely involved in reward has anything to do with adverse effects or toxicity. If the mechanism is involved in these adverse effects, the authors should elaborate to make the connection clear.
10. In the regulation section, I encourage the authors to consider adding a discussion related to the effectiveness of health and wellness resources, including clinics like the Providence Crosstown Clinic in W. Hastings St., Vancouver, CA. This style of treatment greatly contrasts many policies, but will provide another dimension to the authors discussion. Additionally, the authors may consider adding a discussion related to toxicity and deaths in countries that do not implement regulations on illicit drugs (e.g., Uruguay). It may address important questions related to the effects of decriminalization of illicit substances.
Paragraph on L200-206 can use editing.
Author Response

(The authors gave the same response as above.)

Reviewer 3 Report
The manuscript is improved.
Author Response
Dear Reviewer
We greatly value your previous insightful feedback and your review of our work. Thank you very much for making this manuscript is more valuable for readers.
Best regards,
Authors
Round 3
Reviewer 2 Report
This is a second revision of a narrative review focused on drug toxicity and regulation. The authors have done a commendable job once again professionally addressing reviewer concerns. Below are minor suggests for the authors to consider:
1. The authors made a strong attempt at providing a mechanistic explanation for the toxicity caused by illicit substances (section 2. Mechanistic Classification). However, the authors should request to leave this out of the manuscript as it 1) does not directly relate to the narrative in some parts (lines 64-69 are more closely associated with rewarding properties rather than toxicity) and 2) the toxicity-related mechanisms are better suited within each drugs section and often already are, making section 2. superfluous.
2. Section 2, it is not clear how drug toxicity is related to the overactivation of the mesolimbic DA system (lines 64-65). A reference is required. Also, it is suggested that the authors remove, “encouraging pharmacological and natural reward.”
3. Include a reference after the sentence, “chronic overstimulation of CB1 cannabinoid receptor activation…” (line 122-124). Otherwise, this sentence may be removed.
4. Include a reference after the sentence, “This propensity is attributed to TCH acting as a partial agonist for CB1 receptors…”(lines 169-170). Otherwise, this sentence may be removed.
5. Include a reference after the sentence, “It is worth mentioning that cannabis toxicity could…”(lines 170-172) and “Moreover, individuals with a history of heart disease…”(lines 172-173).
6. Include a reference after the sentence, “Plant-derived natural drugs…”(lines 435-436).
7. Based on the references cited (110-113) following the sentence, “Consequently, a prison crisis…”(lines 497-499) the authors may consider replacing this sentence with something less direct. This is because it appears that more evidence is required to determine whether decriminalization results in better outcomes (e.g., harm reduction, reduced rates of drug use, reduced drug-related deaths). Maybe the authors could consider something along the lines of, Although countries have enacted decriminalization policies, evidence for or against its effectiveness is lacking (110-113).”
Author Response
Dear Reviewer,
Thank you very much for your insightful comments. We have revised the article based on what you suggested. Please have a look at the attached files. Thank you
Author
